# Late Preterm Newborns: Breastfeeding and Complementary Feeding Practices

**DOI:** 10.3390/children11040401

**Published:** 2024-03-28

**Authors:** Ieva Dijokienė, Raminta Žemaitienė, Dalia Stonienė

**Affiliations:** Department of Neonatology, Faculty of Medicine, Lithuanian University of Health Sciences, LT-44307 Kaunas, Lithuania; ieva.ankudaviciute@stud.lsmu.lt (I.D.); raminta.bradulskyte@stud.lsmu.lt (R.Ž.)

**Keywords:** late preterm infants, breastfeeding, complementary feeding, baby-friendly hospital initiative

## Abstract

Background: The aim of this study is to identify factors associated with successful breastfeeding in late preterm infants (LPIs) and explore the initiation of complementary feeding; Methods: Prospective cohort study was conducted of infants born at 34^+0^ to 36^+6^ weeks gestational age in the Hospital of the Lithuanian University of Health Sciences Kaunas Clinics during 2020–2021. Families were followed up until the infants reached 12 months of age. Average breastfeeding initial time, average breastfeeding duration time, prevalence of exclusive breastfeeding and average solid-food feeding initiation time were examined. The correlations among factors that might affect breastfeeding rates were calculated using the chi-square test (*p* < 0.05); Results: In our study with 222 eligible participants, we observed a statistically significant delay in breastfeeding initiation only in the 34^+0+6^ gestational age group (*p* < 0.001). At discharge, the 36^+0+6^ group exhibited a significantly higher exclusive breastfeeding rate (*p* < 0.001). Over the first year, breastfeeding rates varied, with no correlation found between duration of exclusive breastfeeding and gestational age. Initial solid-food feeding times were similar across groups, and all infants were introduced to vegetables first; Conclusions: Vaginal delivery, skin-to-skin contact after birth, early rooming-in, and breastfeeding within 2 h after birth statistically significantly causes earlier breastfeeding initiation and longer duration of breastfeeding in LPIs. All infants began solid-food feeding at an average age of 5 months, with vegetables being the primary food choice.

## 1. Introduction

The first hours and days of a newborn are crucial for the successful start of lactation and breastfeeding. The benefits of breast milk for the newborn are well known. The World Health Organization (WHO) recommends that all newborns, including premature, sick or low-birth-weight infants, should be fed with breast milk [1]. Although premature or sick newborns may not be able to suckle immediately, they should still be breastfed over time [2]. The small gestational age of an infant causes increased feeding problems: only 7% of term infants face feeding difficulties compared to 32% of late preterm infants. Insufficient feeding was found to be one of the most common reasons for prolonged hospital stay after birth in late preterm infants [3].

Breastfeeding premature newborns, regardless of where they are cared for, whether with their mother or in a neonatal unit, poses many challenges due to the physiology of suckling and lactation, psychology of the early postpartum mother and environmental pressures on both the mother and the newborn [4]. Non-nutritive sucking was identified to have a significant positive effect on the duration of transition from gavage to full oral feeding [5]. Donor milk is the first choice of food for newborns if its own mother’s breast milk is not available. Fortification is recommended for infants weighing under 1800 g [6]. Systematic reviews showcase the importance of environmental and professional support, skin-to-skin contact (SSC), early transfer of the newborn to the same ward as the mother (rooming-in), timely initiation and maintenance of lactation, early colostrum administration and encouraging milk donation for successful breastfeeding [1].

In 1991, the United Nations Children’s Fund (UNICEF) and WHO launched the Baby-Friendly Hospital Initiative (BFHI) to protect, promote and support breastfeeding as well as benefit mothers and their newborns. The initiative consists of the Ten Steps to Successful Breastfeeding (the Ten Steps), focusing on the well-being of the mother and newborn. While the Ten Steps are easily proven to significantly increase breastfeeding rates in healthy, full-term newborns, there is evidence that they have benefits for preterm newborns too [7]. In 2018, the updated guidelines for the implementation of the BFHI expanded the possibility of interpreting the Ten Steps to include groups of preterm, sick and low-birth-weight newborns. The aim is to help hospitals and medical staff to promote breastfeeding in these groups to accomplish the best possible results for newborns. The support of the medical staff is essential for a successful breastfeeding start. As well as the mother’s ongoing involvement in the newborn’s care, frequent SSC help the mother to get to know her premature or sick newborn better, to know her hunger or stress signals, to gain confidence and to reduce anxiety about the condition of her newborn [1]. 

The aim of this study is to identify factors associated with successful breastfeeding in late preterm infants (LPIs), with the goal of ensuring their optimal development and mitigating potential risks associated with prematurity.

## 2. Materials and Methods

### 2.1. Design and Setting

A prospective cohort study was conducted at the Hospital of the Lithuanian University of Health Sciences, Kaunas Clinics, encompassing infants born within the gestational age range of 34^+0^ to 36^+6^ weeks from 1 September 2020 to 31 August 2021. Comprehensive follow-up procedures were implemented to track the progress of the enrolled infants, and data collection was extended until the infants reached 12 months of age.

The Hospital of the Lithuanian University of Health Sciences Kaunas Clinics is a tertiary level perinatal center with 15 beds in the neonatal intensive care unit (NICU), accredited as a baby-friendly hospital (BFH) in 2004, and offers family-centered care.

Late preterm infants, regardless of gestational age or weight, are admitted to Level I care (well newborn nursery) under the condition of clinical stability and the absence of medical support needs. Infants necessitating ventilatory, cardiovascular, or nutritional support are admitted or transferred to Level II (special care nursery), or Level III (neonatal intensive care) based on their clinical requirements.

### 2.2. Sample

The target population for our study comprised mothers who delivered newborns with a gestational age ranging from 34^+0^ to 36^+6^ weeks at our hospital. Inclusion criteria encompassed infants born within this gestational age range from 34^+0^ to 36^+6^ weeks. All infants meeting these criteria were included upon written parental consent. Nine mothers did not agree to participate in the study, and one infant died in the hospital. A total of 222 infants were enrolled in the study. The study sample is described in detail in Figure 1.

### 2.3. Data Collection

Mothers whose infants met the inclusion criteria were enrolled in the study on the day their infant was discharged. At enrollment, the following maternal variables were collected through medical records: age, mode of delivery (vaginal delivery/caesarean section), and number of previous live births. Neonatal variables were also recorded through medical history, including gestational age, birth weight, gender, Apgar score (skin color, heart rate, reflexes, muscle tone, respiration) at 1 and 5 min, skin-to-skin contact after birth, breastfeeding within 2 h after birth, the need for respiratory support, early rooming-in after birth, initiation of enteral feeding, and mode of feeding at discharge.

Families were followed up until the infants reached 12 months of chronological age.

Mothers were contacted at 1, 3, 6, and 12 months of infant age through phone calls by a single investigator (R.Ž.). Following a structured interview, mothers were asked whether their infant had been exclusively breastfed and/or had been fed with formula. Infants were categorized:Exclusively breastfed (including expressed mother’s milk).Partial breastfeeding included infants that were fed both with breast milk and formula.Exclusively formula-fed infants.

If mothers indicated that their infants had received semi-solid and/or solid food, further inquiries were made regarding the timing of the initial introduction of complementary foods. Specifically, they were asked about the schedule and timing of the introduction, as well as the specific type of food that was first introduced into the infant’s diet. Food types were categorized into distinct groups, including fruits and vegetables, various porridges (such as cow’s milk-curd porridge), and the baby-led weaning method.

During the 12-month phone interview, mothers were specifically questioned about the timing and types of foods that had been introduced into their infants’ diets. Additionally, they were asked about the circumstances and reasons surrounding the weaning of infants from breast milk. The interview further explored maternal experiences related to introducing foods and inquired about any pertinent details regarding the presence of older siblings in the family.

### 2.4. Statistical Analysis

The statistical analysis employed in this study utilized R version 4.2.2 (R Core Team, Vienna, Austria, 2023). Participants were stratified into three sub-groups based on gestational age: 34^+0+6^, 35^+0+6^, and 36^+0+6^. For qualitative data pertaining to variables such as breastfeeding initiation time, timing of weaning from breastfeeding, and timing of solid food introduction, frequencies and percentages were presented. Quantitative data, inclusive of means and standard deviations (SD), were expressed and rounded to one digit above the pooled data.

The examination of associations, particularly those related to the timing of breastfeeding initiation, duration of exclusive breastfeeding and/or complementary food introduction and infant and maternal variables, was conducted through the Chi-square test and univariate linear regression analysis.

All statistical analyses were carried out at a 5% level of significance. *p* values, serving as indicators of statistical significance, were rounded to three decimal places. Instances where *p* values were less than 0.001 were denoted as <0.001, while those exceeding 0.999 were denoted as >0.999.

## 3. Results

A total of 222 infants born at 34^+0^ to 36^+6^ weeks gestational age born to 187 mothers were recruited and completed the study. The basic characteristics of the mother–infant pairs that completed the study are shown in Table 1.

The data of our study suggest variations in the average breastfeeding initiation time among different gestational age groups. There was no statistical significance between singletons and multiplets (33 pairs of twins and a triplet) in breastfeeding initiation and duration; thus, we did not specify them in the data analysis. For the 34+0+6 group, the average initiation time was 5.8 days (SD 4.4). The 35^+0+6^-group exhibited a shorter average initiation time of 2.4 days (SD 2.2). The 36^+0+6^-group had the shortest average initiation time at 1.6 days (SD 2.1). A statistical analysis revealed that the breastfeeding initiation time for the 34^+0+6^ group was significantly delayed compared to the other groups (*p* < 0.001).

The proportions of exclusive and partial breastfeeding varied among gestational age groups at the time of discharge. Specifically: in the 34^+0+6^ group, 33.3% exclusively breastfed and 60.1% partially breastfed. The 35^+0+6^ group showed rates of 36.2% for exclusive breastfeeding and 52.9% for partial breastfeeding. The 36^+0+6^ group had proportions of 51.0% for exclusive breastfeeding and 37.8% for partial breastfeeding.

Importantly, statistical analysis revealed that the 36^+0+6^ group had a significantly higher rate of exclusive breastfeeding at discharge compared to the other groups (*p* < 0.001). This underscores the noteworthy difference in breastfeeding practices at the time of discharge, particularly in the context of varying gestational ages.

The proportions of exclusive breastfeeding, partial breastfeeding, or exclusive formula feeding at the first year of life are presented in Table 2

Breastfeeding rates vary across gestational age groups and different time points in the first year. Group 36^+0+6^ exhibits higher rates of exclusive breastfeeding compared to the other groups at 1 month. However, no statistical significance was observed. The rates of partial breastfeeding generally decrease over time in all gestational age groups, with a notable decline at 12 months.

The mean duration of breastfeeding was examined across different gestational age groups: 34^+0+6^ average duration of breastfeeding was 5.9 months (SD = 4.7), 35^+0+6^ average duration of breastfeeding was 6.9 months (SD = 4.6), and 36^+0+6^ average duration of breastfeeding was 6.5 months (SD = 4.9). Our analysis sought to explore potential correlations between the duration of exclusive breastfeeding and gestational age. However, no statistically significant correlation was identified in our investigation, indicating that the duration of exclusive breastfeeding did not exhibit a significant association with the gestational age of the infants under consideration.

Our study focused on factors that might influence breastfeeding initiation and/or duration rates, as presented in Table 3 and Table 4.

Our study examined the average initiation time of solid-food feeding among infants across three distinct gestational age groups: 34^+0+6^ (mean = 5.5 months, SD = 1), 35^+0+6^ (mean = 5.3 months, SD = 0.9), and 36^+0+6^ (mean = 5.4 months, SD = 0.9). Despite the meticulous analysis, no statistically significant differences in the timing of solid food introduction were observed. Noteworthy is the uniform observation that infants from all three gestational age groups were introduced to vegetables as their inaugural solid meal. The statistical assessment for correlation between gestational age and the choice of first solid meal yielded a *p*-value of 0.512, indicating an absence of statistically significant correlation. Infants, regardless of gestational age, universally began solid-food feeding at an average age of 5 months, with vegetables being the primary food choice.

## 4. Discussion

According to a study in Japan, risk factors for successful breastfeeding include having a primiparous mother aged 35 years or older [8]. In our study, maternal age did not have a significant impact on breastfeeding success, except in the 34^+6^ group, where slightly older mothers (aged 30–39 years) were statistically significantly more likely to breastfeed for at least 3 months. Also, primiparous mothers of 36^+6^ newborns in the study were more likely to breastfeed for more than 6 months. The result of the Japanese study contradicts ours; probably, it is reasonable to conclude that more investigation should be done.

Our study shows that mothers who gave birth naturally tend to continue breastfeeding for longer than those who had a caesarean section. Similar results were found in Canada, where Singh et al. concluded that neonates who were delivered by caesarean have higher chances of infant health and behavior difficulties than neonates who were delivered vaginally; as well, their mothers report low milk supply more often [9]. After vaginal birth, catecholamines reach their peak due to fetal head compression and recurrent hypoxia caused by contractions. These changes keep the neonate alert and aroused during the first 2 h after birth for successful breastfeeding initiation. Li, L et al., in their literature review, concluded that not only do infants born via caesarean have difficulties with initiation of breastfeeding, as well as being breastfed for a shorter duration, but that pain, as physical limitation and lack of support, complicates the breastfeeding [10].

Skin-to-skin contact (SSC) is very beneficial for newborns and mothers. SSC calms the newborn after birth, stabilizes its cardiopulmonary system and improves thermoregulation. SSC also has long-term benefits for the newborn: immediately after birth, SSC helps the newborn’s skin to take over the mother’s non-pathological skin microbiota, which contributes to the development of immunity and resistance to disease, as well as to improved brain development. Timely and continuous SSC facilitates the initiation of breastfeeding [1,11]. The colostrum, which the newborn receives in the first hours after birth, is rich in antibodies and other biologically active substances that are useful and necessary for the newborn [4]. The subsequent SSC, also known as Kangaroo Mother Care (KMC), is highly recommended for both premature and full-term newborns. For the mother, KMC facilitates lactation; stimulation of the skin increases the release of oxytocin, which is responsible for the let-down reflex and the contraction of the milk ducts, facilitating the flow of the milk [11,12]. In low-birth-weight and preterm infants, KMC improves thermoregulation, reduces the risk of nosocomial infections, decreases the incidence of apnea, and improves the survival of low-birth-weight and preterm infants [13,14]. In addition, frequent KMC improves the formation of the mother–newborn dyad bond. In our study, SSC immediately after birth was associated with a more successful initiation of breastfeeding; however, the incidence of subsequent KMC was not assessed, but all mothers are encouraged to KMC their newborns as often as possible, once the newborn’s condition is stable enough.

In 1991 UNICEF, together with WHO, launched the Baby-friendly Hospital Initiative (BFHI)—ten steps to protect, promote and support breastfeeding globally. Step 7 states: “Enable mothers and their infants to remain together and to practice rooming-in 24 h a day” [1,7]. Early rooming-in is standard practice in our hospital when caring for full-term stable newborns. Newborns born vaginally are together with their mothers 24 h a day, if the conditions of the newborn and the mother permit, and newborns are taken out for short periods for vaccinations, check-ups and blood tests, if necessary; even in these cases, the parents are allowed to accompany the child and be present during the procedure. Neonates born by caesarean section are also placed on the mother’s chest after birth for the first SSC and breastfeeding, if possible. Depending on the indications for the operation in the specific situation, the conditions of the mother and the child, the newborn may not be separated from the mother and travels with her to the intensive care unit for post-operative observation and then on to the obstetric ward. Premature newborns, in the absence of a need for NICU treatment, are often monitored by neonatal nurses for some time in the special care nursery to ensure thermoregulation, stable cardiopulmonary function and effective feeding. Parents are always welcome to be present in the special care nursery with their newborns for breastfeeding and KMC. However, it is the early rooming-in of preterm neonates with their mother that our study shows has a significantly positive effect on the initiation and duration of breastfeeding. 

Complementary feeding is a phase that raises many questions for parents. According to WHO guidelines, exclusive breastfeeding is recommended until 6 months of age, and the Lithuanian Dietetic Association recommends earlier introduction of complementary feeding only in certain clinical situations: premature birth, lack of certain nutrients, growth disorder or other health conditions of the infant [15]. In these conditions, the decision to provide earlier complementary feeding should be made on an individual basis. In our study, solid foods were introduced at 5.5 months (22 weeks) of chronological age. Considering the adjusted age, infants born at 34 weeks of GA were introduced to solid foods at 16 weeks. According to ESPGHAN recommendations, complementary feeding is possible from 17 weeks of age, but the recommendations are only given for full-term infants [16]. There is a lack of data on complementary feeding of preterm infants and its impact on the digestive system. The currently popular baby-led weaning (BLW) method has been used in only a few subjects, although studies have shown that it is safe, does not increase the risk of choking and does not cause iron deficiency later in life [17]. The majority (90%) of infants started on a vegetable diet, i.e., a low-calorie, low-protein diet, which is also one of the recommended ways [16,18]. A lot of mothers indicated choosing potato as first vegetable, potatoes being indispensable in the Lithuanian national cuisine.

Although the WHO recommends continuing breastfeeding up to 24 months of age [19], only 30% of infants in the study group were breastfed at 12 months. Consumption of any amount of breast milk at discharge seems to have a protective effect on regaining weight and length catch-up growth at 36 months of chronological age for LPIs and small-for-gestational-age infants [20]. A systematic review by Kramer and Kakuma states that infants exclusively breastfed for at least six months (26 weeks) demonstrate no growth deficits and they contract less severe gastrointestinal infections compared to infants breastfed partially for three or four months [21]. Mothers cited poor weight gain, insufficient milk supply (the infant suckles for a long time and seems hungry), problematic sleep or tiredness of the mother, and inability to devote time to breastfeeding due to older children in the house as reasons for stopping. A few mothers stopped breastfeeding because of their own health conditions, such as the need for surgery and subsequent hospitalizations, the need to start medication, or the mother’s glycaemia, which was difficult to adjust in the case of Type I diabetes mellitus. Mothers of five subjects reported that their infants weaned themselves at 10–11 months of age and did not receive formula, although breast milk or formula is recommended as the main food for infants up to the age of 1 year [19]. 

Many mothers who participated in our study stated that they discontinued breastfeeding because of their infants’ low weight gain. It is commonly understood that formula-fed infants gain more weight; however, this weight gain acceleration in infancy tends to have negative outcomes later in life. According to Juharji H et al., breastfed infants tend to gain more non-fat body mass, which poses a lower risk of childhood obesity [14,22]. Pediatricians and general practitioners caring for infants should be able to make a rational assessment of infants’ weight gain and, if necessary, refer them to a lactation consultant instead of prescribing formula. Perceived insufficient milk supply and actual insufficient milk supply may have no significant correlation according to Galipeau et al., who suggest that interventions in lactation should aim to increase the confidence of the breastfeeding mother, which affects the duration of breastfeeding [23]. Although, Evans, Hilditch and Keir [24] as well as Dib et al. [25] in their systematic reviews came to a conclusion that there is a lack of evidence for what kind of intervention would help to improve breastfeeding rates in LPIs, Estalella et al. performed a quasi-experimental study, which defined that specific support among LPIs, such as promoting parents’ involvement, bringing in a multidisciplinary approach and keeping the mother–infant dyad inseparable, resulted in a higher breastfeeding rate at discharge [26].

## 5. Limitations

We faced some limitations in our study. We did not evaluate the incidence of KMC, which is an important factor in successful breastfeeding.

## 6. Conclusions

Vaginal delivery, skin-to-skin contact after birth, early rooming-in, breastfeeding within 2 h after birth and successful prior breastfeeding statistically significantly causes earlier breastfeeding initiation and longer duration of breastfeeding in LPIs. Infants, regardless of gestational age, universally began complementary food feeding at an average age of 5 months, with vegetables being the primary food choice.

## Figures and Tables

**Figure 1 children-11-00401-f001:**
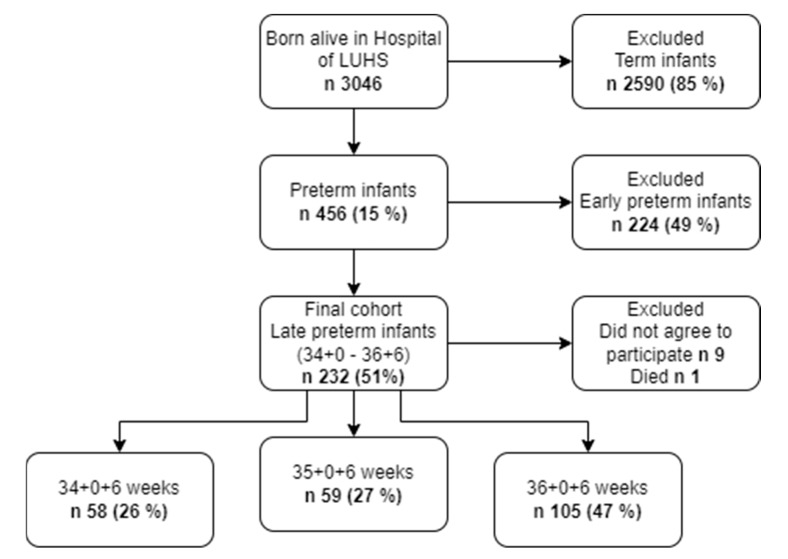
Cohort constitution.

**Table 1 children-11-00401-t001:** Basic characteristics of the mother–infant pairs that completed the study.

Parameter	34^+0+6^	35^+0+6^	36^+0+6^	Total
Mothers	Av. (SD) or % (n)			
Mother’s age	31.1 years (SD 5.1)	31 years (SD 5.9)	31.6 years (SD 5.1)	31.1 years (SD 5.4)
Parity				
Primiparous	37.9% (n = 16)	45.7% (n = 21)	51.7% (n = 45)	43.6% (n = 82)
Multiparous	62.1% (n = 34)	54.3% (n = 30)	48.3% (n = 42)	56.4% (n = 106)
Type of delivery				
Vaginal	56% (n = 28)	56.9% (n = 29)	66.7% (n = 58)	61.2% (n = 115)
Caesarean	44% (n = 22)	43.1% (n = 22)	33.3% (n = 29)	38.8% (n = 73)
Infants	Av. (SD) or % (n)			
Singleton	72.4% (n = 42)	67.8% (n = 40)	67.6% (n = 71)	68.9% (n = 153)
Twins/triplets	27.6% (n = 16)	32.2% (n = 19)	32.4% (n = 34)	31.1% (n = 69)
Birth weight	2331 g (SD 438.8)	2400 g (SD 375.9)	2735 g (SD 461)	2545 g (SD 476.4)
Hypoxia at birth	6.9% (n = 4)	10.2% (n = 6)	2.9% (n = 3)	5.9% (n = 13)
Born without hypoxia	93.1% (n = 54)	89.9% (n = 53)	97.1% (n = 102)	94.1% (n = 209)
Males	62.1% (n = 36)	45.8% (n = 27)	49.5% (n = 52)	61.2% (n = 115)
Respiratory support need	51.7% (n = 30)	35.6% (n = 21)	18.1% (n = 19)	31.5% (n = 70)

**Table 2 children-11-00401-t002:** Breastfeeding practices across gestational age groups.

Type of Feeding	34^+0+6^	35^+0+6^	36^+0+6^	Total
	% (n)			
	1 month of age			
Exclusive breastfeeding	52.5% (n = 32)	48.2% (n = 27)	58.3% (n = 60)	53.6% (n = 119)
Partial breastfeeding	30.5% (n = 17)	37.5% (n = 24)	25.2% (n = 28)	31.1% (n = 69)
Formula feeding	17% (n = 10)	14.3% (n = 8)	16.6 (n = 16)	15.3% (n = 34)
	3 months of age			
Exclusive breastfeeding	37.3% (n = 22)	51.8% (n = 29)	44.7% (n = 46)	43.7% (n = 97)
Partial breastfeeding	25.4% (n = 14)	25% (n = 18)	20.4% (n = 23)	24.8% (n = 55)
Formula feeding	37.3% (n = 22)	23.2% (n = 13)	34.9% (n = 35)	31.5% (n = 70)
	6 months of age			
Exclusive breastfeeding	30.5% (n = 18)	38.2% (n = 21)	35% (n = 36)	33.8% (n = 75)
Partial breastfeeding	18.7% (n = 10)	18.2% (n = 14)	15.5% (n = 18)	18.9% (n = 42)
Formula feeding	50.8% (n = 30)	43.6% (n = 24)	49.5% (n = 51)	47.3% (n = 105)
	12 months of age			
Exclusive breastfeeding	25.9% (n = 15)	34% (n = 18)	28.2% (n = 29)	27.9% (n = 62)
Partial breastfeeding	1.7% (n = 1)	1.9% (n = 1)	4.8% (n = 7)	4.1% (n = 9)
Formula feeding	72.4% (n = 42)	64.2% (n = 40)	67% (n = 69)	68.0% (n = 151)

**Table 3 children-11-00401-t003:** Maternal factors influencing breastfeeding initiation and/or duration rates.

	34^+0+6^	35^+0+6^	36^+0+6^
Mother’s age			
Av. (SD)	31.08 years (SD 5.13)	30.96 years (SD 5.89)	31.57 years (SD 5.06)
Correlation from breastfeeding initiation	No correlation (*p* = 0.22)	No correlation (*p* = 0.948)	No correlation (*p* = 0.48)
Correlation from breastfeeding duration	LPIs born to mothers aged 30–39 years were breastfed more than 3 months, *p* = 0.010	No correlation (*p* = 0.51)	No correlation (*p* = 0.642)
Number of live births			
Correlation from breastfeeding initiation	No correlation (*p* = 0.30)	No correlation (*p* = 0.55)	No correlation (*p* = 0.69)
Correlation from breastfeeding duration	No correlation (*p* > 0.05)	No correlation (*p* > 0.05)	Primiparous breastfed at approximately more than 6 months, *p* = 0.006
Type of delivery			
Correlation from breastfeeding initiation	Earlier breastfeeding occurred among vaginal delivered, *p* = 0.04	No correlation (*p* = 0.169)	No correlation (*p* = 0.186)
Correlation from breastfeeding duration	Vaginally born LPIs breastfed > 3 months, *p* < 0.001	Vaginally born LPIs breastfed > 3 months, *p* < 0.001	Vaginally born LPIs breastfed > 3 months, *p* < 0.001
Breastfeeding duration of older siblings			
Correlation from breastfeeding duration	No correlation (*p* = 0.30)	No correlation (*p* = 0.130)	Longer prior breastfeeding linked to significantly prolonged overall breastfeeding, *p* = 0.000
Prior reasons of weaning			
Correlation from breastfeeding initiation	No correlation (*p* > 0.005)	No correlation (*p* > 0.005)	No correlation (*p* > 0.005)
Correlation from breastfeeding duration	No correlation (*p* > 0.005)	No correlation (*p* > 0.005)	No correlation (*p* > 0.005)

**Table 4 children-11-00401-t004:** Infant factors influencing breastfeeding initiation and/or duration rates.

	34^+0+6^	35^+0+6^	36^+0+6^
Birth weight			
Av. (SD)	2331 g (SD 438.77)	2400 g (SD 375.95)	2735 g (SD 461)
Correlation from breastfeeding initiation	Breastfeeding initiation was earlier in newborns weighing > 2500 g, *p* = 0.032	No correlation (*p* = 0.088)	Breastfeeding initiation was earlier in newborns weighing > 2500 g, *p* = 0.000
Correlation from breastfeeding duration	No correlation (*p* = 0.604)	No correlation (*p* = 0.643)	No correlation (*p* = 0.698)
APGAR score			
Correlation from breastfeeding initiation	No correlation (*p* = 0.296)	No correlation (*p* = 0.078)	No correlation (*p* = 0.355)
Correlation from breastfeeding duration	No correlation (*p* = 0.76)	No correlation (*p* = 0.894)	No correlation (*p* = 0.280)
Skin-to-skin contact after birth			
Correlation from breastfeeding initiation	Skin-to-skin contact longer than 30 min caused earlier breastfeeding initiation, *p* = 0.000	Skin-to-skin contact longer than 30 min caused earlier breastfeeding initiation, *p* = 0.048	Skin-to-skin contact longer than 30 min caused earlier breastfeeding initiation, *p* = 0.01
Correlation from breastfeeding duration	No correlation (*p* = 0.476)	No correlation (*p* = 0.398)	No correlation (*p* = 0.108)
Breastfeeding within 2 h after birth			
Correlation from breastfeeding initiation	Was not performed	LPIs breastfed within 2 h after birth had successful continuation, *p* = 0.038	LPIs breastfed within 2 h after birth had successful continuation, *p* = 0.000
Correlation from breastfeeding duration	Was not performed	No correlation (*p* = 0.063)	No correlation (*p* = 0.066)
Respiratory support need			
Correlation from breastfeeding initiation	No correlation (*p* = 0.130)	No correlation (*p* = 0.096)	LPIs with respiratory distress had delayed breastfeeding, *p* = 0.003
Correlation from breastfeeding duration	No correlation (*p* = 0.30)	No correlation (*p* = 0.130)	No correlation (*p* = 0.280)
Rooming-in after birth			
Av. (SD)	3.96 days (SD 2.83)	2.39 days (SD 2.46)	1.48 days (SD 1.58)
Correlation from breastfeeding initiation	Earlier rooming-in promoted earlier breastfeeding initiation in LPIs, *p* > 0.001	Earlier rooming-in promoted earlier breastfeeding initiation in LPIs, *p* > 0.001	Earlier rooming-in promoted earlier breastfeeding initiation in LPIs, *p* > 0.001
Correlation from breastfeeding duration	LPIs with early rooming-in breastfed for a longer duration, *p* > 0.001	LPIs with early rooming-in breastfed for a longer duration, *p* > 0.001	LPIs with early rooming-in breastfed for a longer duration, *p* > 0.001

All statistical analyses were carried out at a 5% level of significance. *p* values serve as indicators of statistical significance.

## Data Availability

The data presented in this study are publicly available.

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
