# Peer review of "Late Preterm Newborns: Breastfeeding and Complementary Feeding Practices"

_children, 2024, doi:10.3390/children11040401_

Round 1

Reviewer 1 Report

Comments and Suggestions for Authors

This paper investigates factors associated with successful breast feeding in late preterm infants. This is an important topic well presented.

 The study has several strengths:

1.        It investigates a group of infants (late preterm) rather neglected.

2.        The study was well planned, and the participants involved prospectively.

3.        All candidates born between 34+0 and 36+6 were recruited and only few excluded because mothers did not consent, or the infant died. Therefore the results are representative for this specific gestational age group.

4.        Exclusive breast feeding and partial breastfeeding are well defined in advance.

5.        The follow-up lasted 12 months (by telephone interview).

6.        Data regarding introduction of complementary feeding in this gestational age groupe are rare in the literature. The data given in this paper may contribute to establish recommendations for this aspect.   

Suggestions:

-            -  How many twins or triplets were included in the study? Multiple parity may well influence exclusive andded from analysis (as in reference 5) or analysed separately. partial breastfeeding rates. As 222 infants were born to 188 mothers, there must be 34 multiplets, probably mostly twins. Multiplets should be introduced as an additional factor in the analysis and if they differ from singletons should be excluded.

-            -  Give maternal and neonatal data of each gestational age group in a table (age, parity, education, etc; birth weight, sex, twin or singleton, delivery mode, Apgar etc).

Author Response

Thank you for your evaluation. There were no statistical significance between singletons and multiplets (33 pairs of twins and a triplet) in breastfeeding initiation and duration, thus we did not specify them in data analysis. We added additional table for basic characteristics in our revised article.

Reviewer 2 Report

Comments and Suggestions for Authors

Title: "Late Preterm Newborns: Breastfeeding and Complementary Feeding Practices"

This is a prospective study of a cohort of late preterm neonates regarding breastfeeding initiation and complementary feeding practices. It is a single-center study conducted in a Northern European country, with telephone follow-up of a cohort of 222 late preterm infants.

While its conclusions are not groundbreaking regarding factors associated with breastfeeding (vaginal delivery, early initiation, rooming-in, etc.), its significance lies in its exclusive focus on late preterm infants, comparing results by gestational weeks at birth (34, 35, and 36).

Introduction:

Adequate, although it only utilizes four references.

Materials and Methods:

It is well-described, but a table with the initial characteristics of each group (e.g., maternal studies, maternal employment, urban or rural environment, maternal age, parity, type of delivery, need for admission to level II and III units, hospital stay, etc.) is needed. Were there any losses in telephone follow-up? Who conducted the telephone interviews?

Statistics consistently employ "means," although perhaps some analyses would benefit from using "medians." Many figures are presented with two decimal places, which seems excessive given the sample size. Perhaps this aspect should be unified throughout the text to "one decimal place."

Results:

The data on the timing of breastfeeding initiation is strikingly late (are these data correct?).

Line 157 states that the 36-week group shows the highest percentages of exclusive breastfeeding, but the table only shows this for the first month, with the 35-week group having the highest exclusive breastfeeding rates at 35 and 36 weeks.

Table 2 is excessively complex and unclear. It should be improved and perhaps divided into two.

Discussion:

This section could be improved by expanding the bibliography and focusing more on aspects related to late prematurity.

Bibliography:

I believe it is somewhat limited and should be expanded. There are numerous publications on breastfeeding that could enrich the text.

Comments on the Text:

The style and language could be improved. Some words appear in uppercase sometimes and lowercase at others (e.g., "caesarean"). The text needs linguistic revision.

Comments on the Quality of English Language

Although the text is readable in its current format, there are expressions and errors in the text that should be improved upon before publication.

Author Response

Thank for your observations and comments regarding our article.

We expanded the list of references in introduction.

We agree with the suggestion to include a table summarizing essential characteristics of mothers and infants. Not all mothers consented to disclose information about their education and marital status; therefore, we opted to exclude this data. The telephone interviews were conducted by Raminta Žemaitienė, no losses in telephone follow-up, mothers were glad to participate in the study.

Thank you for your recommendation about “means” and “medians” usage, however we have opted for a statistical analysis more suitable with the specifics of our data. We have made the necessary corrections regarding the one-decimal place observation.

When referring to breastfeeding initiation time, we specifically denote exclusive full breastfeeding, with achieved productive suckling.

Other corrections, according to your comments were made directly in the article, as well as expansion of bibliography.

Round 2

Reviewer 2 Report

Comments and Suggestions for Authors

Dears authors,

I appreciate your effort in improving the text based on some of my suggestions. However, I believe that some changes still need to be made before accepting the manuscript:

In the abstract, it is not necessary to include the phrase: "No statistical significance or correlation was observed in this context (p = 0.512)." I recommend removing it.

There is no need to mention in the text the person who made the phone calls. It is only necessary to clarify if it was done by one person consistently or if it was done by more than one person, and if there was any prior training to avoid biases.

In Table 1, it could be interesting for the first column to display data for all cases combined (those at 34, 35, and 36 weeks). The same applies to Table 2.

Table 3 remains unclear. I would recommend dividing it into two or presenting it differently. I apologize for being repetitive, but in an international scientific publication, tables must be meticulously crafted and very clear.

When it was requested in the first review to always use uppercase or lowercase words, it did not mean that "cesarean" had to be capitalized. It is better to always use lowercase for it.

Author Response

Thank you again for your review and helpful suggestions.

“In the abstract, it is not necessary to include the phrase: "No statistical significance or correlation was observed in this context (p = 0.512)." I recommend removing it.”

The phrase was removed.

“There is no need to mention in the text the person who made the phone calls. It is only necessary to clarify if it was done by one person consistently or if it was done by more than one person, and if there was any prior training to avoid biases.”

The corrections were made according to the comment.

“In Table 1, it could be interesting for the first column to display data for all cases combined (those at 34, 35, and 36 weeks). The same applies to Table 2.”

We added column “Total” in both tables.

“Table 3 remains unclear. I would recommend dividing it into two or presenting it differently. I apologize for being repetitive, but in an international scientific publication, tables must be meticulously crafted and very clear.”

Table 3 is indeed very important in displaying the results of our study. We discussed among the authors what is the best way to divide it into two, whether it is where one table describes the influence on breastfeeding initiation and the other the influence on breastfeeding duration, or divide it by influencing factors, e.g. maternal and infant. We chose the latter way. We hope this way it brings more clarity to the results of our study.

“When it was requested in the first review to always use uppercase or lowercase words, it did not mean that "cesarean" had to be capitalized. It is better to always use lowercase for it.”

The corrections were made according to the comment.